# Insulin Resistance, a Risk Factor for Alzheimer’s Disease: Pathological Mechanisms and a New Proposal for a Preventive Therapeutic Approach

**DOI:** 10.3390/biomedicines12081888

**Published:** 2024-08-19

**Authors:** Flora Affuso, Filomena Micillo, Serafino Fazio

**Affiliations:** 1Independent Researcher, Viale Raffaello, 74, 80129 Napoli, Italy; 2UOC of Geriatric Medicine AORN S.G. Moscati, 83100 Avellino, Italy; 3Department of Internal Medicine, School of Medicine, Federico II University of Naples, 80138 Naples, Italy; fazio0502@gmail.com

**Keywords:** insulin resistance, brain insulin resistance, brain metabolism, oxidative stress, cognitive impairment, Alzheimer’s disease

## Abstract

Peripheral insulin resistance (IR) is a well-documented, independent risk factor for the development of type 2 diabetes, cardiovascular disease, cancer and cellular senescence. Recently, the brain has also been identified as an insulin-responsive region, where insulin acts as regulator of the brain metabolism. Despite the clear link between IR and the brain, the exact mechanisms underlying this relationship remain unclear. Therapeutic intervention in patients showing symptoms of neurodegenerative diseases has produced little or no results. It has been demonstrated that insulin resistance plays a significant role in the pathogenesis of neurodegenerative diseases, particularly cognitive decline. Peripheral and brain IR may represent a modifiable state that could be used to prevent major brain disorders. In this review, we will analyse the scientific literature supporting IR as a risk factor for Alzheimer’s disease and suggest some therapeutic strategies to provide a new proposal for the prevention of brain IR and its consequences.

## 1. Introduction

Neurodegenerative diseases (NDs) represent a direct and indirect economic burden whose costs are estimated to double in a few years. Among NDs, Alzheimer’s disease (AD) accounts for 60–80% of cases, representing not only a medical issue, but also an economic and social problem [1]. Along with AD, type 2 diabetes mellitus (T2DM) is rapidly increasing [2]. Given the aging of the population and the close relationship between these conditions, it is imperative to investigate and intervene to reverse the current trend. The reasons for poor outcomes should be sought in the late diagnosis of both conditions and the lack of effective therapies to address the phenomenon. The pathogenesis of AD has many mechanisms in common with T2DM development. Insulin resistance (IR) and aging are crucial points to focus on in order to plan for an effective intervention strategy. To prevent the onset of neurodegenerative alterations due to aging, it is essential to diagnose the condition at an early stage, preferably during the cognitive impairment state or even before. While regarding T2DM, it is crucial to intervene during the IR state, which may precede T2DM by 10–15 years [3]. Despite the lack of a clear understanding of the underlying mechanisms that link these diseases, it has been demonstrated that T2DM is closely related to AD, and numerous common factors have been identified. It is well established that IR represents a recognized feature of both T2DM and AD. The association between IR and the development of poor cognitive performance has been investigated in humans, with several studies demonstrating a clear link between the two conditions [4,5,6]. However, there is also evidence that peripheral IR does not always develop in tandem with central IR [7], which suggests that other mechanisms may also be involved in the pathogenesis of cognitive impairment. In every day clinical practice, not all patients with AD have IR, and the prevalence of IR in these patients is not precisely known. However, a 2004 study published in *Diabetes* shows that both type 2 diabetes (35% vs. 18%, *p* < 0.05) and impaired fasting glucose (IFG) (46% vs. 24%, *p* < 0.01) were more prevalent in AD vs. non-AD control subjects, giving a total of 81% where cases of AD had either type 2 diabetes or IFG [8]. Several factors are involved in the pathogenesis of AD, including inflammation, mitochondrial disfunction, oxidative stress, amyloidosis, genetics, vascular damage, glucose homeostasis alterations and metabolic disorders with or without peripheral insulin metabolic impairment. The aim of this review is to elucidate the known pathophysiological mechanisms of IR and their implication on the neuronal metabolism, which ultimately leads to cognitive disfunction, and to explore the available treatments and propose new insights to improve the therapeutic approach.

To achieve this objective, studies about the pathophysiological mechanisms of neurodegenerative processes and their relationship with IR and T2DM will be reviewed, sourced from Pubmed, Scopus, Science Direct and other relevant databases, using the following keywords: neurodegenerative diseases, cognitive impairment, dementia, Alzheimer’s disease, beta-amyloid, Tau, insulin resistance, insulin, type 2 diabetes mellitus, prevention and treatment of neurodegenerative diseases.

## 2. Insulin, Brain and Insulin Resistance

Insulin is secreted by the pancreas and travels via the bloodstream to the brain, where it crosses the blood–brain barrier (BBB) and is actively transported in a saturable manner [9,10]. Insulin transport into the brain can occur independently of insulin receptors (INSRs), as demonstrated in genetic knock-down endothelial INSRs in mice or the pharmacological inhibition of INSRs, where insulin transport across the BBB was not affected [11,12,13]. It is known that a small amount of insulin, the significance of which is unknown, is directly secreted by various brain districts [14,15]. Brain INSRs have a different structure compared to peripheral receptors [16]. In the brain, insulin binds to INSRs distributed in different regions and drives to a series of phosphorylations that, through the insulin receptor substrates 1 and 2 (IRS-1 e IRS-2), lead to insulin action by activating downstream metabolic pathways. The main kinases involved in insulin signal transduction are phosphatidylinositol 3 kinase (PI3K) and protein kinase B (AKt), which are responsible for neuronal plasticity, survival and neurotransmitter trafficking [17,18,19,20]. Insulin can also activate the mitogen-activated protein kinase (MAPK), which modulates cell growth and proliferation. The MAPK pathway is particularly activated by the insulin growth factor-1 receptor (IGF-1-R), which is more active in this signalling. Notably, brain areas involved in cognition, including the hippocampus, and in the control of peripheral metabolism, such as the hypothalamus, feature a large number of INSRs [21,22]. In addition to the aforementioned cerebral effects, insulin is responsible for cholesterol synthesis, reproduction, olfaction and nutrient homeostasis [23,24]. In the brain, it also acts as an antiapoptotic agent, protecting neurons from oxidative stress, ischemia and βamyloid (Aβ) deposition. It is also involved in the neurotransmitter turnover, particularly increasing dopamine. Insulin and dopamine cross-interact, where insulin increases dopamine uptake while dopamine modulates pancreatic insulin release [25,26,27,28] (Figure 1).

Insulin availability is also regulated by insulin-degrading enzymes (IDEs), which are responsible for insulin degradation in the brain as well as in the periphery. In the BBB, IDEs regulate the amount of insulin reaching the brain and control BBB INSR signalling [29,30]. IDEs have also been identified as regulators of Aβ in neurons and glia [30]. In a clinical study, reduced hippocampal IDE expression was found in patients with mild cognitive impairment (MCI) [31]. Peripheral IR is defined as an inadequate metabolic response to insulin, such that higher levels of the hormones are required to achieve a glucose-lowering response. The same inadequate response has been recognised in the brain, where neuronal cells and cerebral vasculature become poorly responsive to insulin. This can occur for several reasons: low availability of active insulin, impaired cell surface INSR expression or a deficit in downstream signalling [32,33,34]. The high presence of INSRs in the BBB underlines the key role of this structure in insulin action [32,35,36]; high levels of brain insulin or IR can decrease BBB insulin transport, supporting a central feedback mechanism regulating insulin uptake [11,12]. The downregulation of INSRs together with an impaired signal in endothelial cells could reduce insulin uptake. In addition, impaired BBB integrity contributes to the passage of larger amounts of solutes that would otherwise not be allowed through the BBB filter [36,37]. The downstream of the INSR signal is mediated by the IRS, whose impaired phosphorylation cascade is involved in brain IR [32,34] (Figure 2).

Diet, especially a high-fat diet (HFD), has been implicated in brain IR [38]. Increased free fatty acid circulation leads to the release of inflammatory factors, which, as will be discussed later, are responsible for the impairment of the insulin metabolism. Another key point is the cerebral circulation. In fact, disorders of this district have been strongly correlated with the progression of IR and cognitive impairment in patients with T2DM [39]. The presence of both peripheral and brain IR in patients with T2DM results in a more severe deficit in brain insulin signalling than in patients without evidence of peripheral IR [40]. Peripheral hyperinsulinemia may reduce the expression of endothelial BBB INSRs, thereby decreasing permeability to insulin [41]. The common denominator between the peripheral and central IR has been investigated, showing an association between metabolic alterations and cerebral degeneration [42,43,44,45,46,47]. Metabolic diseases have a major impact in the loss of the brain functions, probably due to the oxidative stress [45,46]. Another interesting mechanism has been reported in the literature to explain the altered insulin responsiveness in the brain of AD patients. In fact, some studies have shown that Aβ can aggregate and form soluble oligomers, which are toxic to neuronal synapses. The presence of Aβ oligomers in the brain is intimately related to impaired insulin signalling and has been shown that the presence of Aβ oligomers in the hippocampal region is associated with a reduction in INSRs on the cell membrane, while there is an increase in INSR levels within their cell bodies. This determines a reduced response to insulin, as documented in neurons cultured in presence of Aβ oligomers [48,49] (Figure 2).

## 3. Glucose Metabolism and Dysregulation of Brain Energetics

The brain metabolism is dependent on glucose, which is considered the major energy substrate. In the central nervous system (CNS), glucose is taken up from the blood and transported across the BBB. In contrast to the peripheral metabolism, insulin levels in the CNS have no effect on glucose transport [50,51]. Glucose transporter 1 (GLUT-1), an insulin-insensitive transporter responsible for the uptake of glucose from extracellular fluid, is expressed in the BBB on endothelial cells, astrocytes and microglia. In the brain, GLUT-3 is widely expressed by neurons and glial cells, and its role is to support neuronal cells under different energy demands [52,53,54,55]. The stimulus for GLUT-1 activation is not the glucose level, as in the peripheral metabolism, but the energy demand of the brain for neuronal function. Energy distribution is supported by astrocytes that absorb glucose from blood vessels and deliver it to neurons [56,57]. Unlike GLUT-1 and GLUT-3, GLUT- 4 is the only insulin-sensitive transporter expressed in the brain and is translocated to the plasma membrane upon insulin stimulation [58]. In the periphery, GLUT-4 is the major transporter, whose disruption leads to IR. The role of GLUT-4 in the brain has been partially elucidated by Reno et al. in a mouse model experiment. They suggested a modulation of neuronal glucose sensing and whole-body energy homeostasis, although the exact role is unknown [59]. Dysregulation of brain energetics is responsible for cognitive impairment [60,61]. Hypometabolism is considered a hallmark of AD [62,63,64,65]. It leads to neuronal death, particularly in the hippocampus, cortex and temporal and parietal lobes, regions primarily affected by the disease [66,67,68,69]. During AD and T2DM, transporter function and expression are altered. GLUT-4, especially localised in hippocampus and hypothalamus, plays a role in memory and IR [70,71]. Its expression increases during AD, suggesting an upregulation during brain glucose hypometabolism [72]. In contrast, GLUT-1 is reduced in AD patients, predominantly in the hippocampus and frontal cortex, while GLUT-3 expression is also reduced in the parietal cortex [73,74,75,76,77]. 

## 4. Aβ, Tau, Apolipoprotein E and IR

Aβ and Tau aggregates are the hallmarks of AD. In patients with AD is characteristic the deposition of Aβ as the major component of extracellular plaques, while intracellular accumulation consists of Tau neurofibrillary tangles (NFTs) [78]. Aβ is composed by the amyloid precursor protein (APP), whose proteolysis generates a transmembrane protein localised mainly in the synapses of neurons. Animal studies have shown that T2DM can enhance the development of amyloid plaques and neurodegeneration [79]. Insulin and IGF1 increase the clearance of Aβ in the brain of mice and partially restore Aβ oligomer damage in hippocampus cells line [80], thereby halting the loss of synapses in neurons [81,82]. Tau is an unfolded protein essential in maintaining the stability of microtubules in axons and neurons. Its abnormal phosphorylation leads to microtubule disruption and the release of free Tau molecules that aggregate into NFTs [83]. Tau is hyperphosphorylated by glicogensynthethase kinase 3 beta (GSK3β), a kinase modulated by the PI3K/AKt pathway. Increased activation of GSK3β has been observed during brain IR [84]. Another leading role in AD is played by apolipoprotein E (APOE), which is produced primarily by astrocytes or by neurons under conditions of stress and injury [85,86,87,88,89]. Involved in the lipoprotein and cholesterol metabolism, its physiological function is to maintain and repair neuronal cells and glia. A genetic variant of APOE, namely, APOE4, is responsible for AD onset and progression over time [90]. APOE4 contributes to AD by modifying plaques, probably with a role in Aβ fibrilization and deposition [91,92]. The increased Aβ plaques and the early onset of amyloid pathology have been observed in post-mortem studies [93]. APOE4 bounds to the oligomers of Aβ, stabilising and enhancing the fibrillogenesis [94,95]. In addition, E4 carriers have impaired Aβ clearance [96]. APOE4 is also responsible for increased Tau phosphorylation, independent of the presence of Aβ plaques [97,98]. In the aging brain of APOE4 carriers, cerebral blood flow (CBF) is reduced; in particular, a regional decrease has been observed even in healthy subjects [99,100]. Animal studies have shown that HFD, together with the E4 genotype, induces IR, reducing glucose uptake and blood flow [101]. The presence of E4 may affect the interaction of insulin with its receptors, since, in the presence of this isoform, INSR is confined in endosomes, impairing the binding and downstream signal [102]. APOE4 can cause mitochondrial dysfunction [103], a recognised feature of AD. Not only APOE alleles but also a large number of genetic associations have been identified between IR and AD. Among the most significant, the single-nucleotide polymorphisms (SNPs) of the fat mass and obesity-associated gene (FTO) may lead to obesity, T2DM, metabolic syndrome and a deficit in brain volume, function and cognition [104]. Furthermore, FTO is involved in the ghrelin–IR pathway. Ghrelin is considered an AD biomarker and its changes may play a detrimental role in the association between IR disorders and AD pathogenesis [105]. Other SNPs associated with both AD and T2DM have been identified in the IDE gene and have found to be linked with late onset of AD, irrespective of the presence of APOE4. Many other genetic variants identified may play a role in IR/AD, whose early identification may offer promising tools in the diagnosis and treatment of AD [106].

## 5. Mitochondria, Oxidative Stress, IR and Neuroinflammation

Glucose in the brain is metabolised into adenosine triphosphate (ATP), which is necessary for cellular energy. ATP synthesis is an oxygen-dependent process realised through the oxidative phosphorylation by the enzymatic complex in the mitochondrial electron transport chain (ETC). Mitochondrial disfunction is strictly implicated in NDs, where impaired bioenergetics lead to the increased production of reactive oxygen species (ROS), contributing to disease progression. Mitochondrial disfunction and oxidative stress have been identified as an early feature of T2DM and AD. An impaired mitochondrial number, biogenesis and activity, together with an exaggerated production of ROS, are present in both clinical conditions [107]. In T2DM, there is a large release of proinflammatory cytokines that cross the BBB, causing neuroinflammation and leading to AD [108]. Decreased mitochondrial DNA (mtDNA) and mRNA, encoding mitochondrial genes and the protein of the ETC, lead to the reduced number and size of mitochondria, impaired respiratory chain function, impaired ATP synthesis and the hyperproduction of ROS, which are all characteristics of mitochondrial dysfunction. Mitochondrial biogenesis is controlled by the peroxisome proliferator-activated receptor γ coactivator 1α (PPAR/PGC1α) system. PGC1α regulates the genes involved in the mitochondrial biogenesis and metabolism. The decreased expression of PGC1α has been described in T2DM [109,110] and in the hippocampal cells of AD patients. In AD, Aβ levels are negatively correlated with PGC1α and plaque formation [111]. It has been demonstrated that an increase in PGC1α expression leads to a reinstitution of the respiratory chain and improved insulin sensitivity [112]. Impaired insulin signalling is conducive to mitochondrial disfunction, oxidative stress and the production of advanced glycation end-products (AGEs). IR augments the mitochondrial ROS concentration, thereby enhancing neuronal apoptosis, particularly in the frontal, parietal and temporal lobes. AGEs consist of a non-enzymatic glycated molecules, such as lipids, proteins and nucleic acids, which are responsible for accelerating cellular senescence. AGEs react with oxygen, increasing the production of ROS, oxidative stress and reducing the activity of antioxidant enzymes. Glycation itself is implicated in the amyloid aggregation process. Aβ glycation also correlates with Tau hyperphosphorylation, enhancing the formation of NFTs. Tau glycation leads to the further production of ROS [113,114,115,116]. Mitochondrial autophagy intervenes to limit the increasing ROS damage and ATP depletion. This is necessary to avoid the usual mitochondrial fusion that occurs under energetic stress to maximise energy production, which, in turn, leads to an increase in oxidative phosphorylation [117,118,119]. ROS produced by the mitochondrial oxidative metabolism are involved in several of the following signalling pathways: insulin and nuclear factor k-light-chain-enhancer of activated B cells (NFkB). These are responsible for the transcription of proinflammatory cytokines, such as tumour necrosis factor α (TNFα), interleukin (IL)-1 and IL-6 and c-JUN N-terminal kinase (JNK), involved in the stress response, which, in turn, impairs insulin signalling in the brain [120,121,122,123,124,125,126]. INSR activation inhibits GSK3β [127] but, during IR and T2DM, GSK3β is not inhibited, and its activation stimulates the production of IL-1B, IL-6 and interferon (IFN)ɣ, reducing the production of anti-inflammatory cytokines [128]. Free radicals are physiologically neutralised by an antioxidant response controlled by nuclear factor erythroid 2 (Nfr2), which regulates the activity and the expression of antioxidant enzymes such as superoxide dismutase, glutathione, catalase and H-eme-oxygenase 1 [129]. These are responsible for maintaining cellular redox balance, thus preventing oxidative damage. When this system in equilibrium fails, oxidative stress occurs. Brain IR is associated with oxidative stress and impaired antioxidant function [120,130]. Mice with IR, fed with HFD, showed a positive correlation of homeostatic model assessment-index IR (HOMA-IR) with brain ROS formation and a negative correlation with ATP production [131,132]. Furthermore, ROS generated from damaged mitochondria activate the NLRP3 inflammasome complex, which is responsible for the production IL-1β and IL-18, leading to neuroinflammation. Thus, mitochondrial dysfunction induces inflammation which, in turn, leads to mitochondrial dysfunction [133,134,135]. Mitochondrial proinflammatory molecules act through several ways, the most studied being mtDNA activation of Toll-like receptors, which trigger the microglial inflammatory response [136,137]. Microglia, composed of brain resident macrophages, is an important mediator of inflammation. In AD, it surrounds Aβ plaques, leading to the release of proinflammatory factors such as IL1β and IL6, and TNFα, which initially counteract amyloidosis but, then, became detrimental, leading to neurotoxicity due to chronic activation [138]. 

Some observational studies have indicated that gut microbiota dysbiosis could intervene in the pathophysiologic mechanisms of AD and other NDs. The microbiota–gut–brain axis is a not fully understood bidirectional communication system regulated by very complex and delicate mechanisms. It includes many neural, immune, endocrine and metabolic pathways. Microbiota dysbiosis can increase the permeability of the gut and the BBB, affecting the pathogenesis of the AD and of other NDs, particularly during aging [139]. It seems that the dysbiotic gut microbiome is capable of forming large quantities of amyloid substance and lipopolysaccharides, which, by modulating signalling pathways and stimulating the formation of proinflammatory cytokines, can intervene in the pathogenesis of NDs [140]. The proinflammatory changes induced by gut dysbiosis can trigger both peripheral and cerebral IR, thus becoming a common denominator in initiating and progressing AD.

## 6. Vascular Damage

Vascular damage is present in more than half of AD patients. It is an important contributor to onset and progression of AD [141]. Vascular disfunction is a prelude of Aβ deposition. In conjunction with BBB vascular damage, a recognised prodrome of small vessel disease (SVD) [142], Aβ reduces brain circulation, leading to a shortage of energy substrates and oxygen [143]. A peripheral vascular alteration has been implicated in AD; indeed, in T2DM, hyperglycaemia causes an imbalance in nitric oxide (NO) homeostasis [144] and can double the risk of ischemic stroke [145]. The role of INSRs is fundamental for vasculature balance, where insulin may induce vasodilation by increasing NO through the PI3K/Akt pathway and vasoconstriction by stimulating MAPK, leading to endothelin-1 activation [3]. NO has been proposed as a regulator of vascular reactivity to provide adequate microvascular flow for the metabolic needs of brain cells [146]. Thus, Aβ vascular degeneration with NO imbalance paves the way to AD onset [147]. SVD is responsible for white matter lesions, and is often associated with common vascular risk factors, including T2DM, hypertension, aging, etc. Its radiological features are cerebral parenchymal haemorrhages, lacunar lesions, ischemic stroke, small subcortical infarcts, etc. all recognised causes of cognitive decline [148]. These observations suggest that a large burden of damage may be preventable through the treatment of risk factors and lifestyle intervention.

## 7. Potential Therapeutic Approach

Therapeutic interventions initiated in patients with active AD and based on the hypothesis of amyloid and Tau hyperphosphorylation resulted in very poor outcomes, as did therapies with antioxidant, anti-inflammatory and neuroprotective agents. Even therapies based on the suppression of thiamine deficiency, which is common in AD patients, have produced very conflicting results. More recently, advances in our understanding of the preclinical phase of AD and T2DM have encouraged the search for therapies aimed at preventing neurological changes rather than treating them once damage has stabilised, with a focus on reducing cerebral IR. Currently, drugs known to be potential insulin sensitisers, such as metformin, peroxisome proliferator-activated receptor γ (PPARγ) agonists and insulin mimetic incretin molecules [glucagon-like peptide-1 (GLP-1)], which are insulin secretagogues, sodium–glucose cotransporter inhibitors (SGLT2 Is) and phosphodiesterase-5 inhibitors (Phosd-5 Is), should be considered. It is also important to consider the good results obtained by the intranasal administration of insulin in patients with MCI. Recently, beneficial effects of Phosd-5 Is, used in the treatment of erectile dysfunction and pulmonary arterial hypertension, have been highlighted in the prevention of cognitive disorders and AD. Some natural substances that simultaneously reduce IR and exert anti-inflammatory and powerful antioxidant actions have considerable scientific support in the prevention and treatment of cognitive deficits and AD, although most of the studies are in animals and there are few clinical studies in humans. Among these, berberine, quercetin and L-arginine deserve special attention.

### 7.1. Metformin

The biguanide metformin is a drug with a long history of use in the treatment of T2DM. It is known to have beneficial effects by reducing IR in the target organs of insulin action, reducing fasting insulin levels and improving insulin action in the control of hepatic glucose production [149]. In recent years, it has been reported that metformin crosses the BBB and, given its insulin-sensitising properties, could play a role in improving brain IR and reducing the risk of dementia. A large-scale epidemiological study of 800,000 subjects from the Taiwan National Health Insurance database with a 7-year follow-up showed that the risk of dementia was doubled in diabetic subjects, and that metformin reduced this risk by 35% over 8 years [150]. This observation is supported by a recent study that clearly demonstrated that the discontinuation of metformin in patients with T2DM was associated with an increased incidence of dementia [151]. In addition to the benefits highlighted in diabetic patients, metformin has been shown to reduce the risk of dementia in non-diabetic subjects, probably due to its beneficial action on mitochondrial function and the NDUFA2 gene. The latter is a mitochondrial complex 1-related gene, whose inhibition of expression in the cerebral cortex is associated with a reduced risk of AD and the preservation of cognitive function [152].

### 7.2. PPARɣ Agonists

PPARɣ agonist treatment in diabetic patients has shown to improve target tissue sensitivity to insulin and to reduce Aβ accumulation and neuroinflammation [153,154]. However, the increased incidence of heart failure (HF) in subjects treated with these drugs has significantly limited their use [155].

### 7.3. Glucagon-Like Peptide-1 Receptor Agonists

GLP-1Ras are drugs used to treat patients with T2DM. They can cross the BBB and act in the brain via GLP-1 receptors. Liraglutide has been shown to antagonise neurodegenerative processes and the progression of AD in rat models [156]. It appears that, by counteracting neuroinflammation and oxidative stress, GLP-1RAs reduce the formation of oligomeric Aβ and neuritic plaques and reduce microglial activation, thereby stimulating neurogenesis [157]. A study from a few years ago analysed data from three randomised, double-blind, placebo-controlled trials, involving 15,820 patients with T2DM treated with GLP-1Ras, to test their effect on cardiovascular outcomes. This analysis confirmed that patients randomised to treatment with GLP-1 RAs had a significantly reduced risk of dementia compared to those treated with the placebo [158]. Subsequently, phase 2 and 3 trials were initiated to verify their effects in patients with AD or Parkinson’s disease.

### 7.4. Sodium-Glucose Cotransporter 2 Inhibitors

SGLT2 Is are drugs used for the treatment of T2DM that have demonstrated significant efficacy in reducing cardiovascular events and deaths, even in patients with HF, and have therefore been rapidly incorporated into treatment guidelines for both diabetic and HF patients [159,160]. These drugs have been shown to significantly reduce circulating insulin levels and have a beneficial effect on IR [161]. There is a large body of scientific literature supporting their use also in the prevention of cognitive deficits and AD, although it is insufficient to make definitive decisions [162,163,164,165,166]. A recent study, conducted in a mouse model of T2DM and AD, demonstrated that SGLT2 Is ameliorated IR, while producing a significant improvement of hippocampal-dependent learning, memory and cognitive functions. In addition, hyperphosphorylated Tau levels and Aβ accumulation, were reduced, while brain insulin signalling was increased [162]. Most of the published clinical trials are quite recent. One of them, based on the Taiwan’s National Health Insurance Research database, analysed data on a total of 976,972 patients newly diagnosed with T2DM between 2011 and 2018. After careful analysis, two groups of 103,247 patients each were selected. One was treated with SGLT2 Is and the other was not. The treated group was associated with a significantly (*p* = 0.0021) lower risk of incident dementia compared to the untreated group [163]. Another recent umbrella review and meta-analysis study of 3,046,661 diabetic patients from 100 reviews and 27 cohort/case-control studies, treated with different anti-diabetic drugs, highlighted how treatment with SGLT2 Is, metformin, thiazolidinediones or GLP-1 RAs was associated with a lower risk of dementia [164]. Many other studies [165,166,167] have shown results in the same direction as these commented on, but they are mostly retrospective epidemiological studies, so we are awaiting confirmation from prospective prevention intervention studies, double-blind, randomised and controlled, with a sufficiently long follow-up.

### 7.5. Phosphodiesterase 5 Inhibitors

Phosd-5 Is are approved for the treatment of erectile dysfunction and pulmonary arterial hypertension. By inhibiting phosphodiesterase 5, these drugs enhance the action of NO in all vascular districts, including the brain. It has been highlighted that, by targeting the insulin-sensitive tissues (muscle, adipose tissue, liver tissue and endothelium), Phosd-5 Is can determine an improvement in IR [168]. A study conducted in HFD-conscious mice highlighted how chronic treatment with sildenafil improved the energy balance and insulin action, resulting in lower insulin and fasting blood sugar levels [169]. About two decades ago, an experimental study was published showing how the inhibition of phosphodiesterase-5 improved synaptic function, memory and Aβ load in an AD mouse model [170]. At the same time, tadalafil was shown to improve cognitive function in AD mice by crossing the BBB [171]. Based on the observation that Phosd-5 Is have shown promising results in experimental animal studies as drugs for the treatment of AD, a recent published cohort study of 269,725 men with erectile dysfunction evaluated new cases of AD during a median follow-up of 5.1 years. This study highlighted 1119 new diagnosis of AD and showed that subjects treated with Phosd-5 Is had a lower risk of AD than untreated subjects, and that this risk was lower in subjects who had a greater number of prescriptions [172].

### 7.6. Berberine

There is a large body of scientific literature, albeit in experimental models, demonstrating the efficacy of berberine in the treatment of cognitive disorders and dementia [173,174,175,176]. Berberine is a naturally derived quinolone alkaloid found in many plants. It has been used in traditional Chinese and Indian Ajurvedic medicines for over 2000 years for its many beneficial effects on human health. Among these, one of the best documented is in the treatment of T2DM by reducing IR [177,178]. In recent years, it has also been suggested that berberine may be useful in the prevention and treatment of senile dementia due to its effects on neurotransmitters, inflammation, oxidative stress, the metabolism and other pathways in the brain [179]. The protective effects of berberine in the brain may occur through numerous complexes and not yet fully understood mechanisms. A recent study in human neuronal cells has shown that berberine protects these cells from damage caused by residue 42 of Aβ (Aβ-42). Aβ-42 damages human neuronal cells through the downregulation of circular RNA of istone deacetylase 9 (circHDAC9) and the strong upregulation of microRNa miR-145-5p. The addition of berberine to human neuronal cells reversed this trend, resulting in the protection from Aβ-42-induced damage [180]. Another study in streptozotocin-induced diabetic rats showed that berberine, used as a chronic treatment, lowered blood sugar, reduced oxidative stress and the synthesis of glial fibrillar acidic protein, a marker of astrocytes damage [181] and a promising candidate biomarker for AD [182]. Another study conducted in a transgenic mouse model of AD showed that treatment with berberine, from 2 months to 6 months of age, significantly improved learning deficits, long-term spatial memory and amyloid plaque accumulation compared to mice treated with the control vehicle alone. In particular, the inhibition of GSK3β activity was observed [183]. Another interesting mechanism of action of berberine could be linked to its beneficial action on the microbiota–gut–brain axis. Results from a very recent study show that berberine treatment reduces Aβ plaque formation, mitigates inflammation and improves spatial memory dysfunction in a 5XFAD mouse model (a mouse model encompassing major features of AD), simultaneously alleviating intestinal inflammation, decreasing intestinal permeability and improving the composition of their intestinal microbiota [184]. 

### 7.7. Quercetin

Many studies support the use of quercetin as a protective substance against brain damage and the onset of cognitive deficits and dementia. However, even in this case, these are mostly experimental studies and with few clinical studies [178,185,186,187,188,189,190]. Quercetin is a flavonoid normally found in the daily diet, mainly in vegetables, fruits and tea. It is one of the most potent dietary antioxidants. Some modern pharmacological studies have shown that quercetin can effectively protect the brain from ischemic damage, thanks to its antioxidant, anti-inflammatory and antiapoptotic mechanisms [190]. In addition, quercetin has been shown to improve IR, thus protecting target tissues from dangerous effects of elevated circulating insulin levels [178]. Domoic acid is a marine toxin that causes important neurotoxic changes, particularly in the hippocampus, mainly due to mitochondrial dysfunction and oxidative stress. One study demonstrated how the administration of quercetin to mice treated with domoic acid resulted in a significant improvement in their behavioural performance [187]. Quercetin is classified by the FDA as a senolytic. Senolytics are considered to be the substances that counteract cellular aging. A recent study demonstrated, in a mouse model of AD, how the administration of quercetin ameliorated the Aβ-associated cellular senescence of oligodendrocyte progenitor cells and cognitive deficits [189]. A recent clinical study investigated the effects of quercetin treatment on cognitive function and CBF in 80 healthy men and women aged 60 to 75 years in a randomised, double-blind, placebo-controlled, parallel study with two groups of 40 subjects each. Cognitive function was assessed by the mini mental state examination and CBF by Near-Infrared Spectroscopy at baseline and after 40 weeks of treatment with quercetin or the placebo. The parameters analysed showed a trend towards improvement, albeit without reaching statistical significance. However, it is necessary to take into account the small size of the sample studied and the fact that they were healthy subjects [188]. 

### 7.8. L-Arginine

L-arginine is an amino acid that is converted to NO in the human body. It has been shown to have many beneficial effects in maintaining our well-being. In particular, L-arginine may have a positive effect in reducing insulin resistance [191,192]. Based on its actions, it has been suggested that it may also be useful in the prevention and therapy of cognitive deficits and dementia. The L-arginine/NO pathway has been shown to be dysfunctional in both neurodegenerative and vascular dementia, and the degree of dysfunction is characteristically associated with the levels of neurodegenerative and vascular markers of brain damage, and with the severity of cognitive impairment [193]. A recent study confirmed that alterations in the arginine metabolism may contribute to the pathogenesis of AD [194]. Another study, using a rat model in which AD was induced by hippocampal injections of aluminium, showed that prophylactic intervention with L-arginine, also injected into the hippocampus, protected against AD by increasing NO levels [195]. Agmatine is a metabolite of arginine obtained from its decarboxylation. It is also produced by CNS neurons, as demonstrated in 51 patients with advanced AD, in whom both arginine and agmatine levels were lower than in 31 healthy controls [196]. In a recent study conducted in frail hypertensive elderly patients, 35 subjects were randomised to treatment with L-arginine for 4 weeks and 37 subjects were randomised to the placebo. During the follow-up period, a significant difference in the Montreal Cognitive Assessment test score was found between the two groups, in favour of the patients treated with L-arginine. The results of the evaluation of the protective mechanisms of L-arginine on human endothelial cells were also reported. The cells were incubated with Angiotensin II for 24 h, and L-arginine was also added to some of the cells. The results showed that, in the cells to which arginine was added, the mitochondrial production of ROS was strongly attenuated [197]. It is suggested that the antiaging effects of L-arginine are mediated by increasing the Klotho protein, probably by reducing IR.

### 7.9. Klotho Protein as Therapeutic Target of Neurodegenerative Diseases

Klotho is a transmembrane protein encoded by the Klotho gene, the deficiency of which appears to be at the basis of cellular aging and thus organismal senescence. The Klotho protein, in soluble form, is also secreted into the blood and may act as an endocrine factor. It regulates the activity of many ion channels and growth factors such as insulin and IGF-1. It also protects cells and tissues from damage caused by oxidative stress [198,199,200]. In particular, the βKlotho isoform is highly selective in metabolic tissues. It is involved in the regulation of tissue glucose absorption, bile acid synthesis and the fatty acid metabolism. The relationship between the Klotho protein and IR is not yet fully understood. In fact, there seems to be a relationship only for Klotho levels below 1.24 pg/mcL [201]. It has been shown that Klotho protein levels are inversely correlated with the presence of metabolic syndrome (*p* = 0.013) and negatively associated with abdominal obesity and triglyceride levels [202]. Mice deficient in the Klotho protein show many features that occur during the aging process in humans, such as atherosclerosis, osteoporosis, infertility, hair loss and cognitive decline. They also have a shorter lifespan. In contrast, mice with high levels of Klotho live 30% longer than normal rats and are more resistant to oxidative stress, one of the major insults involved in the aging process [203,204]. There is also a large body of scientific literature linking Klotho protein deficiency and aging to the processes that lead to cognitive deficits and dementia. One study showed that Klotho levels in cerebrospinal fluid were lower in subjects with AD and in older adults than in young adults [205]. Another study highlighted that the upregulation of Klotho protein levels in the brain and serum significantly improved Aβ damage, neuronal and synaptic loss and cognitive deficits in amyloid precursor protein/presenilin mice, to the point that the authors concluded that, given the neuroprotective effects of Klotho overexpression, the latter should be further investigated as a potential therapeutic target for AD [206]. Metformin, SGLT2 Is, berberine, quercetin and probably L-arginine have antiaging actions mediated by their effect on increasing the Klotho protein, probably through beneficial effects on reducing IR [207,208,209,210,211]. 

### 7.10. Inhaled Insulin

As mentioned above, the physiological role of insulin in the brain is not fully understood, but the observed protective effects against the development of AD have made this hormone a candidate for the treatment of cognitive decline. Since peripheral administration has adverse effects, several studies have been attempted with the inhaled intranasal route. In both human and animal studies, intranasal insulin has demonstrated cognitive improvement [212,213]. Acute insulin delivery to the hippocampus in male Sprague–Dawley rats has been shown to improve spatial memory by modulating glucose utilisation in a PI3K manner [214]. A pilot clinical study was performed in 104 adult subjects with AD or mild cognitive dysfunction. In this study, patients were administered placebo or 20 or 40 IU of insulin nasally for a period of 4 months; furthermore, in a small group was also performed a positron emission tomography (PET) study before and at the end of treatment. The results obtained with the tests used indicated a clear improvement in cognitive deficit and delayed memory, while the PET showed an increase in the 18F fluorodeoxyglucose metabolism in most of the brain regions affected before treatment [215]. Unfortunately, however, these studies were not sufficiently strong to give definitive clinical conclusions. This depended on the type of study carried out, but also, partly, on the type of inhaler and the type of insulin used (rapid vs. long-acting), and, in part, on the duration of the treatment [216,217,218,219]. Therefore, a definitive decision on the efficacy of this therapy, also in combination with other drugs used as insulin sensitisers, will have to await the results of ongoing trials that will be completed in a few years [220].

## 8. Conclusions

IR appears to be a key factor in several pathologies, including T2DM and AD. Peripheral and CNS IR do not necessarily occur simultaneously in both districts, but the frequent association between the two forms of IR should be taken into account to explore the common and the distinct underlying mechanisms. As discussed, mitochondrial dysfunction, oxidative stress, inflammation and vascular impairment are all features of IR that are recognised in both diseases. Combined intervention targeting the different dysfunctional processes may improve the outcomes of these diseases. Given the increasing incidence of AD and T2DM, and the lack of effective treatments for AD, it is imperative to intervene in the preclinical phases to counteract the modifiable factors. It would be beneficial to deepen our understanding of brain IR and its relationship with peripheral IR. Large clinical trials should be started as soon as possible to verify whether a preventive therapeutic intervention could avoid or, at least, slow down cognitive impairment before the AD onset. 

## Figures and Tables

**Figure 1 biomedicines-12-01888-f001:**
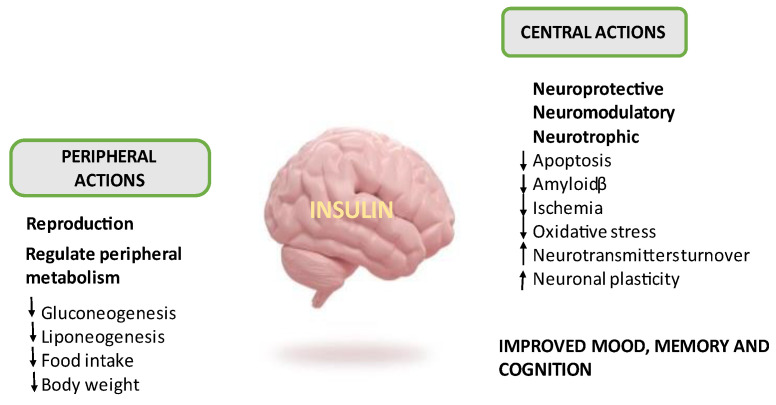
Insulin action in the brain: central and peripheral effects.

**Figure 2 biomedicines-12-01888-f002:**
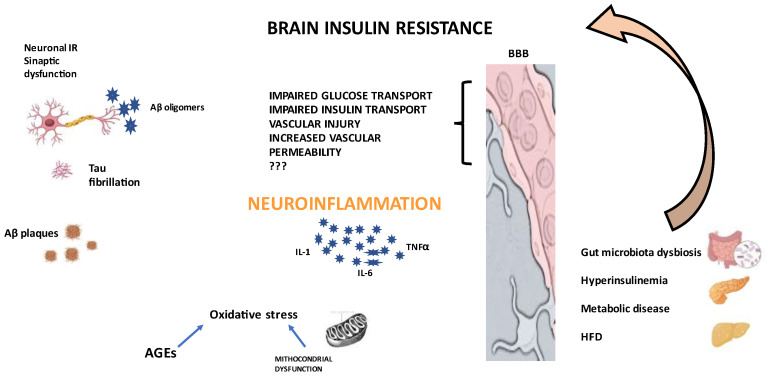
Schematic illustration of brain insulin resistance and pathological hallmarks of Alzheimer’s disease BBB: brain blood barrier; Aβ: amyloid beta; IL-1: interleukin 1; IL-6: interleukin 6; TNFα: tumor necrosis factor α, AGEs: advanced glycation end products, IR: insulin resistance, HFD: high fat diet.

## Data Availability

No new data were created or analyzed in this study. Data sharing is not applicable to this article.

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
