# Peer review of "Insulin Resistance, a Risk Factor for Alzheimer’s Disease: Pathological Mechanisms and a New Proposal for a Preventive Therapeutic Approach"

_biomedicines, 2024, doi:10.3390/biomedicines12081888_

Round 1

Reviewer 1 Report

Comments and Suggestions for Authors

In this study the authors summarized literature supporting insulin resistance (IR) as a risk factor for Alzheimer’s disease and suggested some therapeutic strategies to provide new proposal for the prevention of brain IR and its consequences. Some concerns and suggestions are listed as below:

Figures are too simple (general) in this review and should be improved significantly.

In clinical practice, not all patients with AD have IR.

Genetic factors regarding the relationship between AD and IR should be discussed.

Table(s) can be used to summary related findings.

The authors said that in both human and animal study intranasal insulin demonstrated  cognitive improvement, while detailed information should be provided and discussed.

Comments on the Quality of English Language

fine

Author Response

To Reviewer 1

We thank very much the Reviewer for his precious suggestions. All additions and changes made to the test are written in red.

In this study the authors summarized literature supporting insulin resistance (IR) as a risk factor for Alzheimer’s disease and suggested some therapeutic strategies to provide new proposal for the prevention of brain IR and its consequences. Some concerns and suggestions are listed as below:

1.Reviewer. Figures are too simple (general) in this review and should be improved significantly.

1. Authors. We agree with the Reviewer, but we chose to make simple and schematic figures so as not to make the content of the manuscript boring. However, according to your suggestion we have tried to improve the figures.

2. Reviewer. In clinical practice, not all patients with AD have IR.

2. Authors. Yes, we agree with the Reviewer that not all the patients with AD have IR, however we have cited an important study in which is reported a prevalence of 81% if we sum patients with cognitive deficit or AD.

3. Reviewer. Genetic factors regarding the relationship between AD and IR should be discussed.

3. Authors. Following the Reviewer’s suggestion we have added a short paragraph regarding genetic factors.

4. Reviewer. Table(s) can be used to summary related findings.

4. Authors. If not essential, we would prefer to avoid tables.

5. Reviewer. The authors said that in both human and animal study intranasal insulin demonstrated cognitive improvement, while detailed information should be provided and discussed.

5. Authors. According to the reviewer’s suggestion we have modified this paragraph.

Reviewer 2 Report

Comments and Suggestions for Authors

The manuscript submitted by Affuso, Micillo and Fazio is a review where authors discuss the potential mechanisms by which insulin resistance may contribute to cognitive decline and, especially, in the progression of Alzheimer's Disease. In addition, the authors discuss the potential for pharmacological treatments to target insulin resistance and prevent or slow the progression of dementia.

Overall, the manuscript is a well-written and informative review. The pathology of AD is very complex, with multiple connections to metabolism. The authors have comprehensively reviewed the literature on the connections between IR and Alzheimer's disease. However, short questions are suggested for the authors to consider to make the review more complete.

-       The authors suggest that insulin passes through the BBB via specific receptors (BBB INSRs). However, other authors indicate that it passes passively through pores. The authors should investigate this issue further to clarify how this transport would occur and understand why insulin availability to the CNS is limited. This view about the brain's insulinergic circuit is suggested in case it helps the authors. https://doi.org/10.24875/ciru.18000572

-       In recent years, the microbiota-gut-brain axis has received significant attention in many disorders, including AD and metabolic disorders. Authors could consider including some information about that. In addition, some beneficial effects of berberine may be attributed to an improvement in the microbiota. The authors could review the bibliography and comment on what is true about this and if a connection exists with AD. https://doi.org/10.1016/j.phrs.2020.104722

-       Some authors relate the existence of beta oligomers with the reduction of insulin receptors on the surface of the neuronal membrane. In case it is of interest to the authors to talk about this process: https://doi.org/10.1172/jci64595

Minor comments:

-       The authors should check some errors in the text. Some examples:

o   Line34 It is essential

o   Line 46 Inflammation

o   Line 87 IDEs have

o   Line 117, 382 extra blank space

o   Line 137 In contrast

-       The legend in Figure 1 is off-centre.

-       The figures could be a little more elaborate or add some more about the mechanisms.

Author Response

To Reviewer 2

We thank very much the Reviewer for the time dedicated to us. All additions and changes made to the test are written in red.

  1. Reviewer.The authors suggest that insulin passes through the BBB via specific receptors (BBB INSRs). However, other authors indicate that it passes passively through pores. The authors should investigate this issue further to clarify how this transport would occur and understand why insulin availability to the CNS is limited. This view about the brain's insulinergic circuit is suggested in case it helps the authors. https://doi.org/10.24875/ciru.18000572

1.Authors. We agree with the Reviewer’s suggestion. Therefore, we have added a short paragraph to better clarify this topic.

  1. Reviewer.In recent years, the microbiota-gut-brain axis has received significant attention in many disorders, including AD and metabolic disorders. Authors could consider including some information about that. In addition, some beneficial effects of berberine may be attributed to an improvement in the microbiota. The authors could review the bibliography and comment on what is true about this and if a connection exists with AD. https://doi.org/10.1016/j.phrs.2020.104722

2.Authors. We agree with reviewer and have added a paragraph on this topic in the chapter 5 and where are described the mechanisms of berberine.

  1. Reviewer.Some authors relate the existence of beta oligomers with the reduction of insulin receptors on the surface of the neuronal membrane. In case it is of interest to the authors to talk about this process: …..

3.Authors. According to the Reviewer’s suggestion, we have added some sentences on this important issue.

4.Reviewer. Minor comments:

-       The authors should check some errors in the text. Some examples:

o   Line34 It is essential

o   Line 46 Inflammation

o   Line 87 IDEs have

o   Line 117, 382 extra blank space

o   Line 137 In contrast

-       The legend in Figure 1 is off-centre.

-       The figures could be a little more elaborate or add some more about the mechanisms.

4.Authors. According to the Reviewer’s suggestions we have checked some errors in the text. Furthermore, although we chose to make only simple and schematic figures, so as not to make the manuscript boring, we have changed the two figures trying to improve it.

Round 2

Reviewer 1 Report

Comments and Suggestions for Authors

The authors have addressed my previous concerns.

Comments on the Quality of English Language

fine